# Injection strategy - a driver of atmospheric circulation and ozone response to stratospheric aerosol geoengineering

Ewa M. Bednarz[1,2,3], Amy H. Butler[2], Daniele Visioni[4,5], Yan Zhang[3], Ben Kravitz[6,7], Douglas G. MacMartin[3]

1. Cooperative Institute for Research in Environmental Sciences (CIRES), University of Colorado Boulder, Boulder, CO, USA
2. NOAA Chemical Sciences Laboratory (NOAA CSL), Boulder, CO, USA
3. Sibley School of Mechanical and Aerospace Engineering, Cornell University, Ithaca, NY, USA
4. Department of Earth and Atmospheric Sciences, Cornell University, Ithaca, NY, USA
5. National Center for Atmospheric Research (NCAR), Atmospheric Chemistry Observations and Modelling (ACOM), Boulder, CO, USA
6. Department of Earth and Atmospheric Sciences, Indiana University, Bloomington, IN, USA
7. Atmospheric Sciences and Global Change Division, Pacific Northwest National Laboratory, Richland, WA, USA

*Correspondence to*: Ewa M. Bednarz (ewa.bednarz@noaa.gov)

**Abstract.**

Despite offsetting global mean surface temperature, various studies demonstrated that Stratospheric Aerosol Injection (SAI) could influence the recovery of stratospheric ozone and have important impacts on stratospheric and tropospheric circulation, thereby potentially playing an important role in modulating regional and seasonal climate variability. However, so far most of the assessments of such an approach have come from climate model simulations in which $SO_2$ is injected only in a single location or a set of locations.

Here we use CESM2-WACCM6 SAI simulations under a comprehensive set of SAI strategies achieving the same global mean surface temperature with different locations and/or timing of injections: an equatorial injection, an annual injection of equal amounts of $SO_2$ at 15°N and 15°S, an annual injection of equal amounts of $SO_2$ at 30°N and 30°S, and a polar strategy injecting $SO_2$ at 60°N and 60°S only in spring in each hemisphere.

We demonstrate that despite achieving the same global mean surface temperature, the different strategies result in contrastingly different magnitudes of the aerosol-induced lower stratospheric warming, stratospheric moistening, strengthening of stratospheric polar jets in both hemispheres and changes in the speed of the residual circulation. These impacts tend to maximize under the equatorial injection strategy and become smaller as the aerosols are injected away from the equator into the sub-tropics and higher latitudes. In conjunction with the differences in direct radiative impacts at the surface, these different stratospheric changes drive different impacts on the extratropical modes of variability (Northern and Southern Annular Mode), including important consequences on the northern winter surface climate, as well as on the intensity of tropical tropospheric

Walker and Hadley Circulations, which drive tropical precipitation patterns. Finally, we demonstrate that the choice of injection strategy also plays a first-order role in the future evolution of stratospheric ozone under SAI throughout the globe. Overall, our results contribute to an increased understanding of the fine interplay of various radiative, dynamical and chemical processes driving the atmospheric circulation and ozone response to SAI, as well as lay the ground for designing an optimal SAI strategy that could form a basis of future multi-model intercomparisons.

## 1. Introduction

Stratospheric Aerosol Injection (SAI) is a proposed solar geoengineering method aimed at temporarily offsetting some of the negative impacts of rising greenhouse gas levels and the resulting increases in surface temperatures. The method would involve the injection of sulfate aerosols or their precursors into the lower stratosphere, which would then reflect a portion of the incoming solar radiation in a manner similar to that observed during past explosive volcanic eruptions (e.g. Kravitz and

MacMartin, 2020).

    Many SAI studies focus primarily on the direct impacts of SAI caused by the reduction in the incoming solar radiation on the large-scale temperature or precipitation pattern. However, SAI would not perfectly cancel the GHG-induced temperature changes because the spatial and temporal pattern of the long-wave radiative forcing from GHGs is very different from that of

the reflection of solar radiation by aerosols (e.g. Bala et al., 2008). In addition, the absorption of radiation by stratospheric sulfate will increase temperatures in the lower stratosphere. Both effects can drive important dynamical changes in stratospheric and tropospheric circulation, resulting in potential 'side-effects' on a regional and/or seasonal scale. These include impacts on extratropical modes of variability, namely the Northern Annular Mode (NAM) and North Atlantic Oscillation (NAO) in the Northern Hemisphere (NH) (e.g. Banerjee et al., 2021, Jones et al., 2022), and Southern Annular

Mode (SAM) in the Southern Hemisphere (SH) (e.g. Bednarz et al., 2022a, 2022b). Both of these are tightly coupled to the variability in the winter stratosphere (Thompson and Wallace, 1998, 2000; Baldwin and Dunkerton, 2002), and forced changes in circulation often project onto these modes (e.g. Thompson et al., 2000; Ring and Plumb, 2007). SAI can also drive changes in the tropical tropospheric circulation, including the Hadley Circulation (e.g. Cheng et al., 2021) and Walker Circulation (e.g. Simpson et al., 2019), which drive precipitation patterns over large parts of the tropical and subtropical regions.


    SAI-induced changes in stratospheric composition can also be important. First, stratospheric concentrations of water vapour are primarily driven by the tropical cold point tropopause temperatures as any excess water vapour is dehydrated upon entry into the stratosphere. The SAI-induced increase in lower stratospheric temperatures due to aerosol heating would increase cold point tropopause temperatures and thus allows more water vapour to enter the stratosphere (e.g. Krishnamohan et al., 2019;

Visioni et al., 2021a; Bednarz et al., 2023). In the lower stratosphere, water vapour acts as a greenhouse gas, trapping a portion of outgoing terrestrial radiation and reradiating it back to the surface. This effect has been shown to constitute an important

contribution to the man-made climate warming (Forster and Shine, 1999; Solomon et al., 2010; Banerjee et al., 2019; Nowack et al., 2023), although few studies have estimated the importance of this effect under SAI. Based on three out of four models participating in the GeoMIP G4 experiment, the radiative forcing of aerosol-induced increase in stratospheric water vapour –

0.02-0.35 ppmv at 100 hPa in the tropics - was estimated at 0.004-0.077 $Wm^{-2}$ (Pitari et al., 2014), but estimates under different SAI scenarios and strategies or in more models have so far been missing. Second, SAI will also impact the stratospheric ozone layer via multiple mechanisms, such as an enhancement of heterogeneous halogen activation on sulfate aerosols, and the resulting halogen catalysed chemical ozone loss, as well as via the SAI-induced changes in the large-scale stratospheric transport (e.g. Tilmes et al., 2018a; 2021; 2022). Stratospheric ozone plays an important role in ecosystem and human health

by shielding the surface from harmful UV-B radiation, and so its past and future evolution has been subject to thorough attention due to international accords such as the Montreal Protocol and its subsequent amendments and adjustments. Recently, the impacts of SAI on ozone were addressed in a new chapter of the WMO Ozone Assessment report (WMO, 2022).

Importantly, by far most of the assessment of climate impacts from a hypothetical SAI deployment comes from simulations

that inject $SO_2$ only at a single location, usually at or near the equator (e.g. as it was done in most of the GeoMIP experiments, e.g. Kravitz et al., 2015; Visioni et al., 2021b), or a combination of locations (e.g. 30°N + 15°N + 15°S + 30°S injections, as was done in the recent GLENS and ARISE-SAI CESM simulations, Tilmes et al., 2018b; Richter et al., 2022). Some assessment of the injection latitude dependence for the simulated climate impacts comes from simulations injecting fixed amounts of $SO_2$ at single latitudes (Richter et al., 2017; Tilmes et al., 2018a; Bednarz et al., 2023; Visioni et al., 2023), and

these demonstrate a strong dependence of the simulated atmospheric and surface climate impacts on the latitude of SAI. However, whilst important for understanding mechanisms driving the SAI response, such single-point SAI simulations are unlikely to be representative of a plausible SAI deployment strategy as single hemispheric deployments would strongly impact interhemispheric temperature gradients and, thus, lead to substantial precipitation changes (e.g. Haywood et al., 2013; Visioni et al., 2023). Recent studies exploring results from more than one strategy utilised an atmosphere-only model configuration

and fixed amounts of injections, focusing on only two or three idealised strategies (Franke et al., 2021; Weisenstein et al., 2022), and/or primarily in-situ microphysical and radiative changes in the stratosphere (Laasko et al., 2022).

Rather than using fixed amounts of injections, Zhang et al. (2023) introduced a set of comprehensive SAI strategies that span a larger part of the strategy space, all achieving the same global mean surface temperature through different combinations of

location and timing of $SO_2$ injections and a feedback algorithm to determine the injection amounts. These strategies were simulated in the Community Earth System Model (CESM2-WACCM6) in a fully coupled configuration and included: an annual equatorial injection, an annual injection of equal amounts of $SO_2$ at 15°N and 15°S, an annual injection of equal amounts of $SO_2$ at 30°N and 30°S, and a polar strategy injecting $SO_2$ at 60°N and 60°S only in spring in each hemisphere. The combination of injection latitudes was selected because it was found that depending on the magnitude of global mean cooling,

there is a limited number of different injection strategies capable of yielding significantly different surface climate (Zhang et

al., 2022). We note that unlike the other recent CESM GLENS and ARISE-SAI simulations, these strategies control only global mean surface temperatures (and not their interhemispheric and equator-to-pole gradient),but are easier to replicate across models and as such are better suited for larger inter-model comparisons. Zhang et al. (2023) proved that despite achieving the same global mean surface temperatures, these four different injection strategies mentioned above lead to different impacts on regional surface climate. Here we use the same simulations as Zhang et al. (2023) but examine impacts on aspects of the climate system that previously have not been explored in detail in relation to injection strategy, including stratospheric climate and chemistry (such as water vapour and ozone) and atmospheric circulation. The latter includes impacts on the stratospheric Brewer Dobson Circulation and modes of extratropical variability, with resulting impacts on the mid-/high latitude surface climate, and on Hadley and Walker Circulations, which drive regional tropical precipitation patterns.

## 2. Methods

### 2.1 SAI strategies and model simulations

We use the CESM2-WACCM6 Earth system model (Gettelman et al., 2019; Danabasoglu et al., 2020) with interactive ocean and sea-ice, interactive modal aerosol microphysics (MAM4, Liu et al., 2016) and interactive middle atmosphere chemistry (MA, Davis et al., 2023).. The horizontal resolution is 1.25° longitude by 0.9° latitude, with 70 vertical levels in hybrid-pressure coordinate up to ~140 km. The model configuration simulates internally generated Quasi-Biennial Oscillation, albeit with generally too weak amplitude in the lower stratosphere and not extending sufficiently low down (Gettelman et al., 2019).

We use the set of SAI simulations introduced by Zhang et al. (2023). The Coupled Model Intercomparison Project Phase 6 (CMIP6) Shared Socioeconomic Pathway SSP2-4.5 experiment is chosen as the background emission scenario. As discussed in MacMartin et al. (2022), this scenario is roughly consistent with the Paris Agreement's Nationally Determined Contributions without increased ambition (Burgess et al., 2020; UNEP, 2021). Four SAI strategies are used, each consisting of three ensemble members and covering the period 2035 to 2069 inclusive. These employ a feedback algorithm that adjusts $SO_2$ injection rates in order to maintain the global mean surface temperatures at the baseline level, corresponding to 1.0°C above preindustrial conditions (henceforth referred to as BASE_1.0) and in CESM2 historical and SSP2-4.5 runs corresponds to the mean over the period 2008-2027 (see MacMartin et al. 2022, for details on the definition of climate targets).

The first strategy, henceforth denoted 'EQ', injects $SO_2$ at the equator at 21.5 km at a constant rate throughout any given year. In the second and third strategy, denoted '15N+15S' and '30N+30S', $SO_2$ is injected at 21.5 km at a constant rate throughout a year and at equal rates at a pair of injection latitudes - 15°N and 15°S or 30°N and 30°S, respectively. The fourth strategy, denoted 'POLAR', injects $SO_2$ at 15 km with equal rates at both 60°N and 60°S but only during spring at each hemisphere (so MAM for 60°N and SON for 60°S). Averaged over the last 20-years of the simulations, the injection rates needed to maintain the global mean near-surface air temperatures at 1.0C above preindustrial levels are 21.0, 16.3, 14.4 and 20.4 Tg-$SO_2$/yr for

EQ, 15+15, 30+30 and POLAR, respectively (Zhang et al. 2023). In all cases, we analyse the last 20-years of the simulations - i.e. 2050-2069 - and, with the exception of the results in Section 6, compare them to their baseline period - i.e. 2008-2027.

## 2.2. The simulated sulfate aerosol fields

The characteristics of the stratospheric circulation imply that the overall distribution of sulfate aerosols strongly depends on the location of the injection. In particular, the equatorial 'EQ' strategy and, to a lesser extent, the tropical '15N+15S' strategy lead to significant confinement of aerosols inside the tropical pipe and thus, to the maximum aerosol mass mixing ratios at low latitudes (Fig. 1a and Fig. S1). The subtropical '30N+30S' strategy, where $SO_2$ is injected largely outside the tropical pipe, allow for more transport of aerosols to the mid- and high latitudes, leading to more uniform aerosol distribution (Fig. S1c). This is in agreement with the results of Weisenstein et al (2022) who compared a 2-point 30°N+30°S injection strategy with a strategy injecting $SO_2$ uniformly over the whole tropics using three independent chemistry-climate models, showing more uniform aerosol distributions if the injections occur in the subtropics. Finally, the 60°N and 60°S injections in POLAR result in the aerosol concentrations maximizing in the high latitudes, with only very small levels in the tropics (Fig. 1b). Notably, the lifetime of aerosols decreases as $SO_2$ is injected away from the equator and into the mid- and high latitudes (Visioni et al., 2023), leading to smaller average sizes of sulfate aerosols ((Fig. 1c-d and Fig. S1) and, thus, larger sulfate surface area densities (Fig. -1e-f and S1. This can be explained by the stronger confinement of aerosols inside the tropical pipe for tropical injections favouring condensational growth over coagulation due to higher concentrations in gridboxes where continuous injections are happening (Visioni et al., 2018; 2020)

Since stratospheric circulation varies seasonally, the simulated aerosol fields under annual injections (EQ, 15N+15S, and 30N+30S) also exhibit a seasonal cycle in their optical depths (Fig. 1e-f and Fig. S1). These changes are, however, much smaller than the seasonal cycle in aerosol optical depths simulated in POLAR, where $SO_2$ is injected only in spring at each hemisphere. The ~1 month timescale for $SO_2$-to-aerosol conversion alongside the seasonality in the OH supply lead to the largest concentrations of sulfate in POLAR simulated in the summer in each hemisphere, before the aerosols are rapidly removed over the following season (see also Lee et al. 2021).

## 3. Annual mean changes in stratospheric climate

### 3.1 Temperatures

Figure 2 (top) shows the yearly mean changes in atmospheric temperatures simulated across the different strategies compared to the baseline period. In particular, Fig.2a-b shows the vertical cross-sections of the responses in EQ and POLAR (which correspond to two extreme ends of our potential strategy space, see Section 2.2), with the analogous responses in 15N+15S and 30N+30S shown in Supplementary Material (Fig. S2, top). The seasonal mean responses are provided in Fig. S4 and are generally qualitatively similar to the yearly mean responses (Fig. 2, Fig. S2); however, seasonal differences will be further

discussed in Section 4. Panel (c) in Fig. 2 shows the yearly mean temperature changes in the tropical lower stratosphere in

each of the four strategies (20°S-20°N, 50 hPa).

We find a strong dependency of the magnitude of the tropical lower stratospheric heating on the SAI strategy, with EQ showing the strongest warming of ~8.8 K at 50 hPa (20°S-20°N) and POLAR showing the smallest warming of ~0.4 K in that region. This can be explained by the spatial distribution of the simulated aerosol cloud, i.e. the amount of sulfate in the tropical lower

stratosphere (Section 2, Fig. 1a-b and Fig. S1), as well as the average aerosol size (with largest, hence more absorptive, aerosols simulated in EQ and smallest, hence less absorptive, aerosols in POLAR; Fig. 1c-d and S1). The results also indicate some strategy dependence of the poleward extent of the lower stratospheric warming, although this is more difficult to isolate as the extratropical stratospheric temperature responses are also strongly controlled by dynamical processes.

### 3.2 Water vapour

The magnitude of the SAI-induced tropical lower stratospheric warming controls cold point tropopause temperatures and is thus directly related to the associated changes in stratospheric water vapour. As shown in Fig. 2d-f, all SAI strategies increase concentrations of water vapour in the stratosphere, with the magnitude of this stratospheric moistening generally tracking the magnitude of the lower stratospheric warming and, thus, SAI strategy. As water vapour in the lower stratosphere acts as a greenhouse gas to warm the troposphere, this secondary effect thus offsets some of the direct tropospheric cooling from the

reflection of solar radiation by sulfate aerosols. The particularly strong increase in lower stratospheric water vapour in EQ, up to 75% at 70 hPa, thus contributes to the low efficacy of this strategy (with 21 Tg-$SO_2$/yr needed in EQ to reach the temperature target, compared to 14 and 16 Tg-$SO_2$/yr in 30N+30S and 15N+15S, respectively; Section 2.1) that is also caused by the strong tropical confinement of aerosols and their larger size (as discussed in more detail in Section 2.2 here and in Zhang et al., 2023).

Regarding the corresponding radiative forcing (RF) from stratospheric moistening, an increase in 1 ppm stratospheric water vapour has been estimated to contribute a radiative forcing of 0.22-0.29 W/m$^2$ at the tropopause (Forster and Shine, 1999; Solomon et al., 2010; Banerjee et al., 2019). We can estimate the radiative forcing from the SAI-induced stratospheric moistening using the simulated changes in water vapour at 100 hPa, i.e. near the cold point tropopause and where radiative effects of stratospheric water vapour on surface temperatures are particularly strong (e.g. Riese et al., 2012), and assuming a

RF of 0.25 W/m$^2$ per 1 ppm $H_2O$ increase. This gives a RF of 0.54 W/m$^2$ in EQ (2.2 ppm increase in $H_2O$ at 100 hPa), 0.31 W/m$^2$ in 15N+15S (1.2 ppm $H_2O$ increase), 0.16 W/m$^2$ in 30N+30S (0.7 ppm $H_2O$ increase) and 0.15 W/m$^2$ in POLAR (0.6 ppm $H_2O$ increase). This can be compared with the RF of GHGs that is being offset, which for the SSP2-4.5 scenario equates to 1.42 W/m$^2$ difference between 2050-2069 and 2008-2027 (Fricko et al., 2017), or with the RF of sulfate itself. For the latter, using the of the top-of-the-atmosphere (TOA) calculations of RF from sulfate from Visioni et al. (2022), whereby a change of

0.588 AOD over 2090-2099 equated to -7.30 W/m$^2$ at TOA, and the global and annual mean sAOD changes in Fig. S1(bottom) here, we can estimate the TOA RF of sulfate in these experiments ranging from -2.90 W/m$^2$ in EQ to -1.43 W/m$^2$ in POLAR.

Hence, whilst the RF from the SAI-induced stratospheric moistening is much smaller than the direct RF of sulfate, it is nonetheless an important contributor and varies strongly across the different strategies; it thus needs to be factored in any considerations of efficacies of individual strategies.

### 3.3. Stratospheric large-scale circulation

### 3.3.1 Zonal winds

Figure 3a-c shows the SAI  yearly mean zonal winds responses for different injection locations (seasonal mean impacts are further discussed in Section 4). The SAI-induced warming in the tropical lower stratosphere drives an anomalous strengthening of the  equator-to-pole meridional temperature gradients near the tropopause and lower stratosphere. This drives an anomalous increase of the subtropical to extratropical stratospheric westerly winds in both hemispheres via thermal wind balance in all seasons and most injection strategies, though more intermittently for the seasonal injection in POLAR (Fig. S4-S5). In the winter and spring hemisphere, especially in the NH, the strengthening of the polar stratospheric jet at ~60° latitude is likely the result of the associated modulation of atmospheric wave propagation and convergence due to the more westerly subtropical winds (Fig. S5; see also e.g. Walz et al., 2023). In accord with the strong dependence of the magnitude of the lower stratospheric warming on the SAI strategy (Fig. 2c), as well as its poleward extent, we also find a strong strategy dependence of the magnitude of the resulting westerly wind increase, with EQ showing the strongest annual mean westerly responses in both hemispheres and POLAR the smallest (e.g. 7 ms$^{-1}$ and 1 ms$^{-1}$ at 50°S and 30 hPa for EQ and POLAR, respectively). For POLAR, the smallest magnitude of the NH polar westerly wind increase may also be partially caused by the increased tropospheric source of wave activity in that strategy, as illustrated by the increased poleward meridional heat flux (V'TH') in the northern midlatitude troposphere (Fig. 3d-f). An increased vertical flux of tropospheric wave activity into the stratosphere could be driven by the enhanced Arctic surface cooling in this strategy and the associated overcompensation of Arctic sea-ice extent in POLAR compared to the baseline period (Fig. S6; see also Zhang et al., 2023). Changes in Arctic sea-ice can drive important changes in wave propagation into the stratosphere and thus vortex strength and temperature (Scinocca et al., 2009; Sun et al., 2014; England et al., 2018); here this effect may thus contribute to the smallest magnitude of the northern polar vortex strengthening in POLAR.

Interestingly, in all four strategies the SAI-induced polar stratospheric westerly response is larger in the SH than in the NH (e.g. 7 ms$^{-1}$ in the SH and 3 ms$^{-1}$ in the NH at 50 hPa for EQ). This could be because of the associated reduction in poleward heat flux simulated in the SH across the three strategies injecting SO$_2$ in the tropics/subtropics, i.e. EQ, 15+15, 30+30 (Fig. 3d-f) for all seasons (not shown). The SAI-induced reduction in tropospheric wave flux propagating into the stratosphere may help to strengthen the austral winter and spring stratospheric westerly response in the SH (Fig. S5), which are the seasons when the stratospheric climatological winds are westerly and wave propagation can occur.  Accordingly, smaller magnitude SH westerly winds responses are found during austral summer (Fig. S4), when responses largely reflect increases due to  the

thermal wind response alone. The reduction of tropospheric wave source in the SH in these three strategies could be because of the associated reduction in the meridional temperature gradients in the SH troposphere and the resulting changes in tropospheric baroclinicity (Fig. 2a-b and Fig. S2 and S4; see also Butler et al., 2010).

### 3.3.2. Brewer Dobson Circulation

The SAI-induced warming in the tropical lower stratosphere and the associated zonal wind changes modulate wave propagation (as discussed in the previous section) and, thus, drive changes in the strength of the large-scale residual circulation. Figure 4 shows the associated changes in Transformed Eulerian Mean vertical and meridional velocities. The increase in the static stability of the troposphere as the result of lower stratospheric warming and the associated reduction in wave breaking in the lowermost stratosphere (illustrated by the enhanced Eliassen-Palm (EP) flux divergence, Fig, S8) weakens upwelling in the upper troposphere and lowermost stratosphere (UTLS) region (Fig. 4a-c), also sometimes referred to as the shallow branch of the Brewer Dobson Circulation (BDC), e.g.: Abalos et al. (2021). By mass continuity, the reduction in vertical velocities is associated with reductions in meridional velocities at the same levels (Fig. 4d-f). In agreement with the strong dependence of the magnitude of lower stratospheric warming on the SAI strategy (Section 3.1.), we find a strong correlation between the magnitude of the SAI-induced weakening of the circulation in that region (Fig. 4c), with the largest changes in EQ and smallest in POLAR.

Warming in the lower stratosphere also reduces the stability of the stratosphere above it and enhances wave breaking in the middle and upper stratosphere, especially in the NH (reduction in the EP flux divergence, Fig. S8), thereby accelerating the deep branch of the BDC (Fig. 4). As with the shallow branch, we find a strong dependence of the magnitude of the response on the SAI strategy. However, there are now more spatial differences in the responses across strategies (Fig. S7), with the upwelling increasing mainly near the latitude of injection. In general, SAI-induced changes in residual circulation in both its shallow and deep branches will drive important changes in stratospheric transport of chemical species, including ozone (Section 6) and water vapour (Section 3.2), thereby modulating their distributions. It will also feedback on the simulated distributions of sulfate itself (e.g. Visioni et al., 2020).

## 4. Seasonal changes in the high latitude circulation and climate

### 4.1 Northern Hemisphere

In the NH winter, changes in the strength of the stratospheric polar vortex can propagate down to the troposphere, affecting the distribution of sea-level pressure and latitudinal shifts of the eddy-driven tropospheric jets. This coupling plays an important role in determining winter temperature and precipitation patterns across the mid- and high latitudes (Thomson and Wallace, 1998). Figure 5 shows the DJF changes in sea-level pressure (a-c), zonal winds at 850 hPa (d-f) and near-surface air temperatures (g-i) simulated in the NH in the different SAI strategies. The NAO sea-level pressure index is calculated as the

difference in sea-level pressure in the Atlantic sector between the midlatitudes (280°E-360°E, 30°N-60°N) and the polar cap (70°N-90°N, all longitudes). The Pacific sea-level pressure index is defined as the sea-level pressure in the Aleutian low region (140°E-240°E, 30°N-70°N). The position of the eddy driven jet is defined as the latitude of the maximum zonal wind at 850 hPa; this is done separately for the Atlantic (280°E-360°E) and Pacific (140°E-240°E) sectors. Finally, near-surface air temperatures over Eurasia and Alaska are defined as the average over 50°N-85°N, 20°E-120°E and over 45°N-70°N, 180°E-

240°E, respectively.

### 4.1.1. Atlantic sector

We find that the SAI-induced strengthening of the stratospheric polar vortex (Fig. 3a-c) only propagates down to the surface and leads to a NAO-like signature for EQ and 15N+15S (Fig. 5a-c and Fig. S9), with the 15N+1S response being weaker than in EQ and not statistically significant. The positive NAO response in these strategies is associated with a poleward shift of the

eddy-driven jet in the Atlantic sector (Fig. d-f5 and Fig. S9), as well as a significant warming in the northern Eurasian region (Fig. 5g-i and Fig. S9). A small cooling also occurs in these SAI strategies in western Europe (Fig. S10). In contrast, no clear NAO-signature or jet shift is found over the Atlantic sector for 30N+30S and POLAR, with the 30N+30S showing only a suggestion of a small equatorward shift of the North Atlantic jet stream instead. We note that all 4 strategies show a poleward jet shift near the jet exit region over Europe (right panel in Fig. S10), and that response is opposite sign to the changes occurring

over the jet entrance region in the west Atlantic sector under 30S+30N and POLAR.

The lack of strong changes in DJF sea level pressure or tropospheric jet shift over the Atlantic sector in POLAR is consistent with the absence of any substantial strengthening of the NH stratospheric polar vortex (Fig. 3b). Despite that, the POLAR strategy does show a marked cooling in the northern Eurasian region, as well as more broadly across much of the Arctic. We

hypothesize that this cooling is a manifestation of the radiative effects from the overcompensation of Arctic sea-ice in that strategy compared to the baseline period (Fig. S6; also Zhang et al. 2023). Increased sea ice would lead to higher surface albedo and greater longwave cooling of the polar surface (e.g. Lee et al., 2023).

### 4.1.2. Pacific sector

In the Pacific sector, EQ, 15N+15S and 30N+30S all show weakening of the sea-level pressure over the north Pacific (Fig. 5a-

c) – corresponding to the strengthening of the Aleutian low – as well as an equatorward shift of the winter jet in that region (Fig 5d-f). The weakening of the Pacific sea-level pressure increases in magnitude and moves slightly poleward as the aerosol precursors are injected away from the equator and more in the subtropics (see. Fig. S9). The SAI-induced strengthening of the Aleutian low gives rise to stronger northerly advection of tropical air on the eastern flank of the anomalous low, increasing temperatures over the western US and Alaskan region (Fig. 5g-i). Stronger northerly advection in the eastern Pacific also

brings in more moist air, and the stronger southerly advection in the western Pacific brings in more dry air; although we note that in these simulations the resulting statistically significant precipitation changes are located mainly above the ocean regions

(Fig. S11). Unlike the dynamically-driven top-down response from the stratosphere in the Atlantic sector, the Pacific response is likely driven directly in the troposphere. Strengthening of the Aleutian low, typically accompanied also by anomalously higher pressures over Canada and anomalously low pressures over the southeast US, is suggestive of a large-scale planetary wave response, typical of wave trains forced by changes to tropical Pacific convective heating associated with the El Niño-Southern Oscillation (ENSO). The Pacific-North American sea-level pressure response is thus likely indicative of large-scale teleconnections from the SAI response in the tropical troposphere, in particular the El-Nino like response in the Pacific (see Section 5). As in the Atlantic sector, POLAR does not show strong changes in sea level pressure or jet shift over the Pacific sector.

## 4.2. Southern Hemisphere

The propagation of stratospheric wind anomalies down to the surface is also an important driver of year-to-year climate variability in the SH, but unlike in the NH where the stratosphere-troposphere coupling occurs primarily in winter, this coupling generally maximizes in austral spring and summer. Figure 6 shows DJF changes in sea-level pressures (a-c)and zonal wind at 850 hPa (d-f) simulated in the SH in the different strategies. We find that both the EQ and POLAR strategies show pattern of sea-level pressure consisting of increased pressure in the mid-latitudes and decreased pressure over the Antarctic. This corresponds to the positive phase of Southern Annular Mode (SAM; here defined as the difference in zonal mean sea-level pressure between 50°S and 70°S), and is accompanied by a statistically significant poleward shift of the SH eddy-driven jet. The magnitudes of these responses are very similar between the EQ and POLAR strategies.

Interestingly, the magnitude of these responses decreases under 15N+15S relative to EQ and POLAR, and the response changes sign under 30N+30S which shows a negative phase of SAM and a small equatorward shift of eddy-driven jet instead. Bednarz et al. (2022b) analysed the SAM changes under fixed single point $SO_2$ injections imposed between 30°S and 30°N in the same CESM2 version, and showed that the SAM response becomes negative under $SO_2$ injections in the SH as the injections are moved further into the subtropics. That work suggested that this occurs because of the poleward extent of lower stratospheric heating impacting planetary wave propagation in the stratosphere as well as eddy heat and momentum fluxes in the troposphere below. It is thus plausible that the SAM and jet responses in the EQ, 15N+15S and 30N+30S strategies here are largely dynamically driven by the lower stratospheric heating, in a manner consistent with Bednarz et al. (2022b). The SH high latitude responses in POLAR on the other hand, where the SAI direct impact is largely focused in the mid- and high latitudes in austral summer (Fig. 1, bottom), is likely primarily driven by the cooling of the Antarctic region caused by the reduced summer insolation under SAI and the subsequent changes in meridional heat transport (in a manner analogous to that inferred for the Arctic in Lee et al., 2023), thereby forcing changes in the SH tropospheric winds and sea level pressures. However, specially designed simulations would be needed to fully diagnose such a mechanism.

We note that a qualitatively and quantitatively similar results regarding the overall behaviour of the SAM and SH eddy-driven jet are obtained if annual mean changes are examined instead of the austral summer responses (Fig. 6c,f).

## 5. Impacts on tropical tropospheric circulation

While changes in global mean precipitation are expected to scale largely, albeit not fully, with the corresponding changes in global mean temperatures (Niemeier et al., 2013; Zhang et al., 2023), any regional SAI-induced precipitation responses will be in part related to changes in the intensity and position of large-scale tropospheric circulation patterns, including the tropical Hadley and Walker Circulations.

Figure 7 shows the simulated yearly mean changes in (c-e) meridional and (f-h) zonal mass stream function. Shown also in (a-b) are the corresponding changes in yearly mean precipitation under EQ and POLAR for reference. As it was done for instance in Guo et al. (2018), we define meridional and zonal mass stream functions ($\varphi_m$ and $\varphi_z$) according to Eq. (1) and (2), respectively:

$$\varphi_m = \frac{2\pi a \cos(\phi)}{g} \int_0^p V \, dp \quad (Eq. 1)$$

$$\varphi_z = \frac{2\pi a}{g} \int_0^p U_D \, dp \quad (Eq. 2)$$

where a is Earth radius, g is the gravitational acceleration, $\varphi$ is latitude, V is meridional wind, and $U_D$ is the divergent component of zonal wind averaged between 10°S-10°N.

We use four metrics of tropospheric circulation. The intensity of the Hadley Circulation is calculated as the difference in $\varphi_m$ at 500 hPa between the northern (10°N-25°N) and southern (25°S-0°) cells. The intensity of the Walker Circulation is calculated similarly but using the difference in $\varphi_z$ at 400 hPa between the Pacific (180°E-240°E) and Indian Ocean (60°E-120°E) cells. The position of the Intertropical Convergence Zone (ITCZ) is defined as the latitude near the equator where $\varphi_m$ at 500 hPa changes sign. The position of the upwelling branch of the Walker Circulation, i.e. the transition between its Pacific and Indian Ocean cells, is defined as the longitude between 80°E-200°E where $\varphi_z$ at 400 hPa changes sign.

We note that a different commonly used metric of the intensity of Walker Circulation used in the literature (e.g. Kang et. al. 2020) is the difference in sea-level pressure between the East Pacific (160°W to 80°W, 5°S to 5°N) and the Indian Ocean/west Pacific (80°E to 160°E, 5°S to 5°N) regions. Since the use of this metric leads to slightly different results we also include this metric in Fig. 6 (circles and dashed lines).

## 5.1. Intensity changes

We find a strong strategy dependence of the simulated changes in the intensity of the Hadley and Walker Circulations. For the Hadley Circulation, EQ shows the largest weakening of the Hadley Circulation, followed by 15N+15S. In contrast, the two other strategies - 30N+30S and POLAR – do not show significant changes in the Hadley Circulation strength in the annual mean. For the Walker Circulation, weakening of its intensity is simulated under all 4 SAI strategies, with larger responses found for the tropical and subtropical injections (EQ, 15N+15S and 30N+30S) and the weakest response in POLAR. The

stronger weakening in EQ and 15N+15S coincides with stronger near-surface air temperature decreases over the Maritime Continent and increases over the eastern Pacific (Fig. S14) indicative of a canonical El-Nino like response; this in turn coincides with more strongly suppressed convection over the western Pacific basin and relatively enhanced convection over the eastern Pacific Ocean (Fig. 7a-b, Fig. S13). In POLAR, near-surface air temperatures increase more broadly over the equatorial Pacific (Fig. S14) with weaker enhancement of convective precipitation over eastern Pacific (Fig. 7a-b), suggestive

of more central Pacific type El-Nino like response (e.g. Yeh et al., 2009). We note that increased convective precipitation in the different El Nino regions can drive different teleconnections patterns (e.g. Calvo et al., 2017), and this effect likely contributes to the strengthening of the Aleutian low simulated in the Northern Pacific under EQ, 15N+15S and 30N+30S in boreal winter (Section 4.1.2).

In general, a weakening of tropospheric Hadley and Walker Circulations corresponds to a weakening of the climatological precipitation patterns, such that the regions characterised by climatologically high precipitation receive anomalously less precipitation, and the regions characterised by the climatologically lower precipitation receive anomalously more precipitation (i.e. dry gets wetter and wet gets drier). The results show that the tropical tropospheric circulation weakens most under strategies injecting $SO_2$ in the tropics and least under high latitude strategy. This behaviour likely arises because of the

combination of how much cooling occurs in each strategy in the tropical troposphere (compared to higher latitudes) as well as to the strength and meridional extent of lower stratospheric heating. The latter increases tropospheric static stability and thus reduces tropical convection, thus adding on to the decrease in the intensity brought about by the purely thermodynamic considerations. See Zhang et al. (2023) for further discussion.

## 5.2. Position changes

In contrast to the strong strategy dependence of the SAI impacts on the intensity of tropical tropospheric circulation, little strategy dependence is found for the changes in the position of the Hadley and Walker Circulations. All SAI strategies show a southward shift of the ITCZ (as also shown in Zhang et al., 2023) as well as an eastward shift of the Walker Circulation. These changes are likely related to the corresponding GHG-induced changes in these metrics under the control SSP2-4.5 scenario (grey points in Fig. 7e,h) and their imperfect compensation under SAI in general. An exception is a stronger eastward shift of

the Walker Circulation under 30N+30S compared to the other strategies; this drives a significant decrease in precipitation over

the Maritime Continent and Australia, and increased precipitation over a large part of the equatorial Pacific (Fig. S13). The stronger eastward shift of the Walker Circulation in 30N+30S also corresponds to larger weakening of its intensity as measured by the sea-level pressure method (points and dashed lines in the bottom right panel in Fig. 7e,h). The stronger response of the Walker Circulation position under 30N+30S compared to other strategies may not be robust (the error bars overlap); regardless, the causes for these different responses are not yet fully understood and remain to be explored.

## 6. Impacts on stratospheric ozone

Figure 8 shows the annual mean changes in ozone mixing ratios and total ozone columns for each of the four injection strategies. In this case we use a comparison against the control SSP2-4.5 simulation during the same period (2050-2069), as opposed to a past period with similar global mean surface temperatures as was done in other sections; this avoids complications from the concurrent long-term changes in ozone caused by long-term changes in ozone depleting substances and greenhouse gases. This approach allows for a better isolation of the impact from increased stratospheric sulfate aerosol load.

### 6.1. Tropics

The increase in lower stratospheric temperatures (Section 3.1) and the resulting weakening of upwelling in the upper troposphere and lower stratosphere (Section 3.3.2) increases ozone concentrations in the tropical lower stratosphere as less ozone poor air is transported from the troposphere. We find a strong strategy dependence of the magnitude of this ozone response, in line with the strong strategy dependence of the SAI-induced changes in lower stratospheric temperatures and transport. A strong correlation between SAI-induced impacts on lower stratospheric temperatures, ozone and transport was also previously reported from other studies, e.g. Bednarz et al. (2023), Tilmes et al. (2022). SAI-induced changes in ozone in the middle stratosphere, in turn, reflect the combination of the SAI-induced strengthening of the deep part of the BDC (Fig. 4), decreasing and increasing ozone levels below and above ozone climatological maximum, respectively, as well as the reduction in active nitrogen species ($NO_x$) upon enhanced heterogenous $N_2O_5$ hydrolysis on aerosol surfaces (Fig. S16); the latter decelerates the gas-phase catalytic ozone loss in the middle stratosphere. Both impacts depend strongly on the SAI strategy, with the largest middle stratosphere BDC and $NO_x$ changes found for EQ and only very small changes for POLAR. As a result of these different competing factors, when integrated over the depth of the atmosphere, EQ shows an SAI-induced total column ozone loss of ~10 DU near the equator (as the ozone decrease at ~30 hPa outweighs the ozone increase below), while no significant tropical total column ozone changes are simulated under the other SAI strategies in the yearly mean.

### 6.2. Northern Hemisphere

The EQ and 15N+15S strategies increase total column ozone in the NH mid and high latitudes throughout the year (Fig. 8-9) - 18 DU and 8 DU on average between 30°N-90°N in yearly mean for EQ and 15N+15S, respectively - because of the SAI-induced changes in ozone transport (Section 3.3.2) and nitrogen mediated gas phase ozone chemistry. In the middle

stratosphere, the acceleration of the BDC brings more ozone from its photochemical production region (i.e. tropical mid-stratosphere) to higher latitudes. In addition, the reduction of $NO_x$ species upon enhanced $N_2O_5$ hydrolysis on sulfate slows down gas phase ozone loss. In the lower stratosphere, on the other hand, the SAI-induced weakening of the shallow branch of the BDC increases extratropical ozone levels as less ozone-poor air is brought from the tropical lower stratosphere.

In contrast, 30N+30S and POLAR show no strong yearly mean total column ozone changes in the NH mid and high latitudes. This occurs because the dynamically-driven ozone increases are much smaller (as the lower stratospheric warming and the resulting BDC changes are weaker), and so are the associated $NO_x$ changes in the middle stratosphere (Fig. S16); these processes are generally offset by the halogen catalysed ozone decreases in the lowermost stratosphere due to the enhanced heterogeneous halogen activation on sulfate (Fig. S16). The halogen mediated ozone losses can however dominate during parts of the year, leading to small ozone reductions (up to ~10 DU) at mid and higher latitudes in the NH spring (30N+30S) and summer (30N+30S and POLAR), Fig. 9.

### 6.3. Southern Hemisphere

In the SH, SAI-induced ozone changes in the mid and high latitudes are even more complex and reflect competing impacts from: (i) enhancement of heterogenous halogen activation on sulfate aerosols and thus increased chemical ozone loss in the lower stratosphere, (ii) strengthening of the Antarctic polar vortex under tropical lower stratospheric warming, reducing Antarctic lower stratospheric temperatures and thus facilitating enhanced formation of supercooled ternary solutions (STS) and polar stratospheric cloud (PSC) and, ultimately, enhancing halogen catalysed ozone loss, as well as reducing in-mixing of ozone-rich mid latitude air into the hight latitudes, (iii) enhancement of $N_2O_5$ hydrolysis and, thus, reduction in chemical ozone loss in the middle stratosphere, and (iv) enhancement of transport of ozone-rich air in the middle stratosphere and reduction of transport of ozone-poor air in the lowermost stratosphere under the SAI-induced BDC changes.

As the result of these competing factors, ozone columns increase under EQ and 15N+15S in the SH subtropics but decrease slightly or stay roughly unchanged for 30N+30S and POLAR. In the mid and high latitudes, all four SAI strategies show significant annual mean column ozone decreases (13 DU, 14 DU, 22 DU and 18 DU ozone loss over 65S-90S for EQ, 15N+15S, 30N+30S and POLAR, respectively, Fig. 8), albeit with differences in magnitudes of these losses between the strategies in different seasons (Fig. 9). While the enhancement of sulfate aerosol surface area densities in the Antarctic stratosphere, leading to the enhancement of halogen activation on sulfate, is strongest for the 30N+30S and POLAR strategies (Fig. 1 and S1), the two tropical injections - EQ and 15N+15S - also show significant Antarctic ozone losses in the lowermost stratosphere over large parts of the year. These are likely driven by the combination of enhanced STS and PSC formation inside colder and stronger polar vortex (enhancing halogen activation, Fig. S16) , as well as by the reduction in in-mixing of the ozone-rich midlatitude air.. Apart from impacting UV transmittance, these lower stratospheric ozone reductions also markedly reduce tropospheric ozone concentrations simulated in the SH mid and high latitudes (Fig. 8 and S15), as less

stratospheric ozone is brought down to the troposphere, with potential consequences for the aerosol cooling efficiency, tropospheric chemistry and air quality.

The results thus highlight the complex interplay of dynamical, chemical and radiative processes driving the stratospheric ozone response to SAI, and call for more research done in quantifying the contributions of individual drivers as well as in narrowing the associated uncertainties, in particular in a multi-model framework.

## 7. Summary and discussion

Most of the assessment of atmospheric and climate response from a hypothetical Stratospheric Aerosol Injection to date comes from climate model simulations in which $SO_2$ is injected only in a single location or a combination of locations. Here we use CESM2-WACCM6 SAI simulations under a comprehensive set of SAI strategies introduced by Zhang et al. (2023) that achieve the same global mean surface temperature with different locations and/or timing of injections: an equatorial injection, an annual injection of equal amounts of $SO_2$ at 15°N and 15°S, an annual injection of equal amounts of $SO_2$ at 30°N and 30°S, and a polar strategy injecting $SO_2$ at 60°N and 60°S only in spring in each hemisphere. Building on the initial results in Zhang et al. (2023), we demonstrate that despite achieving the same global mean surface temperature, the different strategies result in contrastingly different impacts on stratospheric temperatures, water vapour, ozone and the large scale stratospheric and tropospheric circulation, with important implications for the surface climate.

First, the absorption of a portion of outgoing terrestrial and incoming solar radiation by sulfate increases lower stratospheric temperatures. A strong SAI strategy dependence is found for the magnitude of the tropical lower stratospheric warming - ranging from 8.8 K at 50 hPa in the equatorial to 0.4 K in the polar strategy – driven by the differences in the simulated spatial distribution of sulfate aerosols as well as their size. This in turn drives a strong strategy dependence of the resulting stratospheric moistening, with 49% (2.2 ppm) increase in tropical lowermost stratospheric water vapour at 100 hPa in EQ compared to 14% (0.6 ppm) in POLAR. The strong increase in lower stratospheric water vapour constitutes a positive radiative forcing (here estimated at 0.15-0.54 W/m$^2$ depending on strategy) that offsets some of the direct aerosol-induced surface cooling from reduced insolation, thereby constituting an important contributing factor when considering the efficacy of each SAI strategy, as discussed in Zhang et al. (2023). The strong SAI strategy dependence of the lower stratospheric warming also gives rise to a strong strategy dependence of the magnitude of the SAI-induced changes in stratospheric transport, including the increase in stratospheric westerly winds in both hemispheres, the deceleration of tropical upwelling and meridional velocities in the UTLS and shallow branch of the Brewer Dobson circulation, and the acceleration of the deep branch of the residual circulation.

Second, despite clear relationship between injection strategy - and thus the concentration of sulfate in the tropical lower stratosphere - and stratospheric circulation and climate on annual timescales, a more complicated picture emerges regarding SAI impacts on the modes of extratropical variability. In the NH during winter, the descent of the stratospheric westerly response down to the surface in the form of a positive phase of the North Atlantic Oscillation in sea-level pressure and a poleward shift of the eddy-driven jet in the Atlantic sector is only simulated under the two tropical injection strategies, i.e. EQ

and 15N+15S. This leads to a significant warming of near-surface winter air temperatures in the northern Eurasia in these two strategies (as well as a suggestion of a small cooling in western Europe). In contrast, 30N+30S and POLAR do not show a clear SAI-induced responses in these regions as the stratospheric westerly responses are smaller. In the Pacific sector, on the other hand, the three tropical/subtropical strategies – i.e. EQ, 15N+15S and 30N+30S – lead to a strengthening of the Aleutian low in the northern Pacific (somewhat stronger for the subtropical injections) and an equatorward eddy-driven jet shift in that

region.   This increases near surface air temperatures over the western US and Alaskan regions. Unlike the top-down response in the Atlantic sector, the Pacific sector response likely reflects a large scale wave response caused by tropospheric teleconnections and changes in tropical Pacific convective heating, although further work and more idealised experiments would be needed to fully diagnose the details of such teleconnections.

In the SH high latitudes, we find that both EQ and POLAR drive a positive Southern Annular Mode sea-level pressure response alongside a poleward shift of SH eddy-driven jet. The response weakens for 15N+15S and changes signs for 30N-30S, where a small negative SAM sea-level pressure pattern and an equatorward shift of eddy-driven jet is found instead. We suggested differences in the primary driver of the responses between the annual and polar strategies, with the response in EQ, 15N+15S and 30N+30S driven primarily by the influence of lower stratospheric heating and its poleward extent, in a manner consistent

with that proposed in Bednarz et al. (2022b), and the response in POLAR driven mostly by the direct radiative impact of the high latitude cooling.

      Third, the study finds a strong strategy dependence of the SAI-induced changes in the intensity of the tropical Hadley and Walker Circulations in the troposphere. Both EQ and 15N+15S show significant weakening of the Hadley Circulation that is

not reproduced for 30N+30S and POLAR. The Walker Circulation, on the other hand, weakens under all SAI strategies, but the magnitude of the changes is strongest for the tropical and subtropical injections. The stronger weakening of the Walker Circulation in these strategies coincides with near-air surface temperatures increasing preferentially over eastern Pacific resembling a canonical El Nino response; whilst the smaller response in POLAR coincides with more extensive warming over the equatorial Pacific. In general, the SAI-induced weakening of the tropical circulations result in consistent 'dry gets wetter,

wet gets drier' impacts on the tropical precipitation patterns. Unlike a clear strategy dependence of the intensity of tropical circulations, no clear strategy dependence is found for the position of the Hadley and Walker Circulations, with the simulated changes likely indicating imperfect compensation of the GHG-induced changes.

Finally, the results show contrasting SAI-induced ozone responses across the four strategies, and varying depending on the region. In the tropics, SAI-induced ozone changes reflect changes in ozone transport and reductions in nitrogen-mediated chemical loss in the middle stratosphere. These largely cancel out when integrated over the depth of the atmosphere, with significant response found only for the EQ strategy (10 DU ozone decrease at the equator compared to the same period in SSP2-4.5). In the NH mid- and high latitudes, EQ and 15N+15S show year-round column ozone increases driven by the SAI-induced changes in the large-scale ozone transport and the reductions in nitrogen catalysed chemical loss cycles. For 30N+30S and POLAR, on the other hand, these changes are smaller and offset by the chemical ozone losses from the enhanced halogen activation on sulfate, leading to small negative to zero changes in total ozone columns. In the SH, while ozone columns increase in the subtropics for EQ and 15N+15S, in the mid and high latitudes all strategies show reductions in total column ozone (ranging from 13-22 DU over the Antarctic in the annual mean). These are likely driven by the combination of processes, including an enhancement of chemical ozone loss from halogen activation - either on sulfate itself or on increased STSs and PSCs concentrations inside stronger and colder polar vortex - and a reduction in ozone transport inside the strengthened polar vortex, the contribution of which varies depending on the SAI strategy used. Our results thus underscore the need for more research in quantifying the contributions of individual drivers of SAI ozone response as well as in narrowing the associated uncertainties, in particular in a multi-model framework.

More broadly, the results highlight the complex interplay of various radiative, dynamical and chemical processes driving the atmospheric response to SAI, not only on the global but in particular on regional and/or seasonal scales. Since all of these are affected differently by different location of $SO_2$ injection, the study demonstrates the importance the choice of the injection strategy has for the simulated climate outcomes. We have demonstrated that some of the undesirable side-effects of SAI that have been well established for tropical injections - e.g. strengthening of the NH polar vortex and the resulting positive NAO-like surface response in winter, or weakening of the intensity of the Hadley and Walker circulations - appear to be mitigated for extra-tropical and polar injections. However, additional impacts for these strategies, like enhanced halogen activation on sulfate, changes to SAM or strengthening of the large-scale equator-to-pole gradient in case of the latter (see Fig. 1c in Zhang et al. 2023), need to also be considered, highlighting the complexity and trade-offs in evaluating which strategy is most optimal. Given the large uncertainty in model representation of the various contributing processes, the results underscore the need for more research done in narrowing the associated uncertainties, including the evaluation of the strategy dependence in a multi-model framework. An improved understanding of the role of injection strategy from this study is thus particularly relevant for designing and informing future inter-model comparisons, including the next phase of the GeoMIP project.

**Acknowledgements**

We would like to acknowledge high-performance computing support from Cheyenne (https://doi.org/10.5065/D6RX99HX) provided by NCAR's Computational and Information Systems Laboratory, sponsored by the National Science Foundation.

Support was provided by the NOAA cooperative agreement NA22OAR4320151, NOAA Earth's Radiative Budget initiative, Atkinson Center for a Sustainability at Cornell University through SilverLining's Safe Climate Research Initiative, and by the National Science Foundation through agreement CBET-2038246. Support for BK was provided by the National Science Foundation through agreement SES-1754740, NOAA's Climate Program Office, Earth's Radiation Budget (ERB) (Grant
NA22OAR4310479), and the Indiana University Environmental Resilience Institute.

## Authors contributions

EMB analysed the results and wrote the manuscript. YZ and EMB carried out the simulations, with assistance from DV. All authors contributed to the discussion of the results and writing of the manuscript.

## Competing interests

The authors declare no conflict of interests.

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

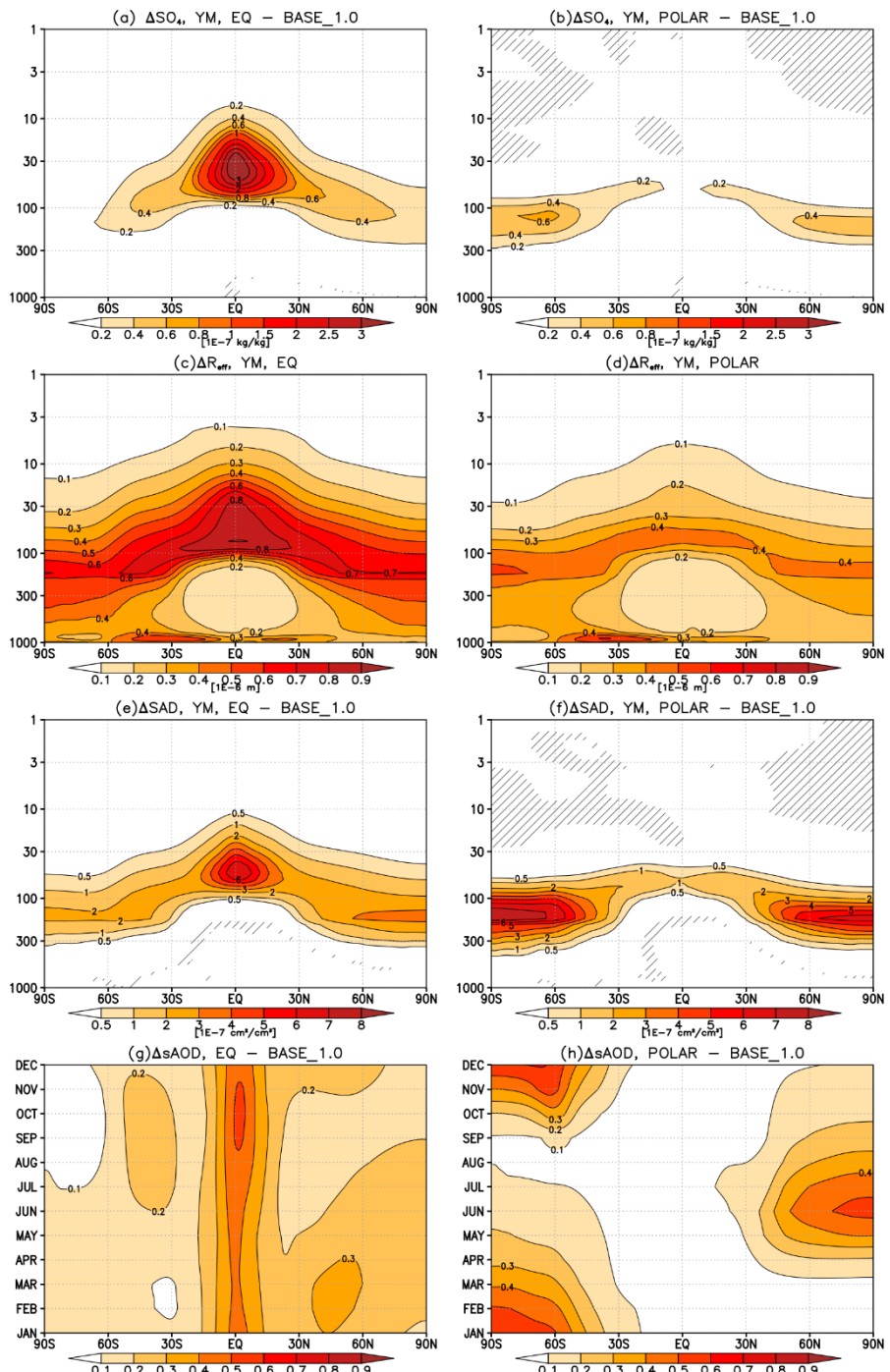

**Figure 1 (Top)** Yearly mean changes in sulfate mass mixing ratios, (row 2) yearly mean aerosol effective radius, (row 3) yearly mean changes in sulfate surface area density, and (bottom) spatiotemporal evolution of simulated AOD changes in EQ and POLAR strategy compared, if applicable, to the baseline period (2008-2027). See Fig. S1 in Supplementary Material for the analogous changes in the 15N+15S and 30N+30S strategies. Hatching in rows 1, 3, 4 indicate regions where the response is not statistically significant, taken as smaller than ±2 standard errors of the difference in means.

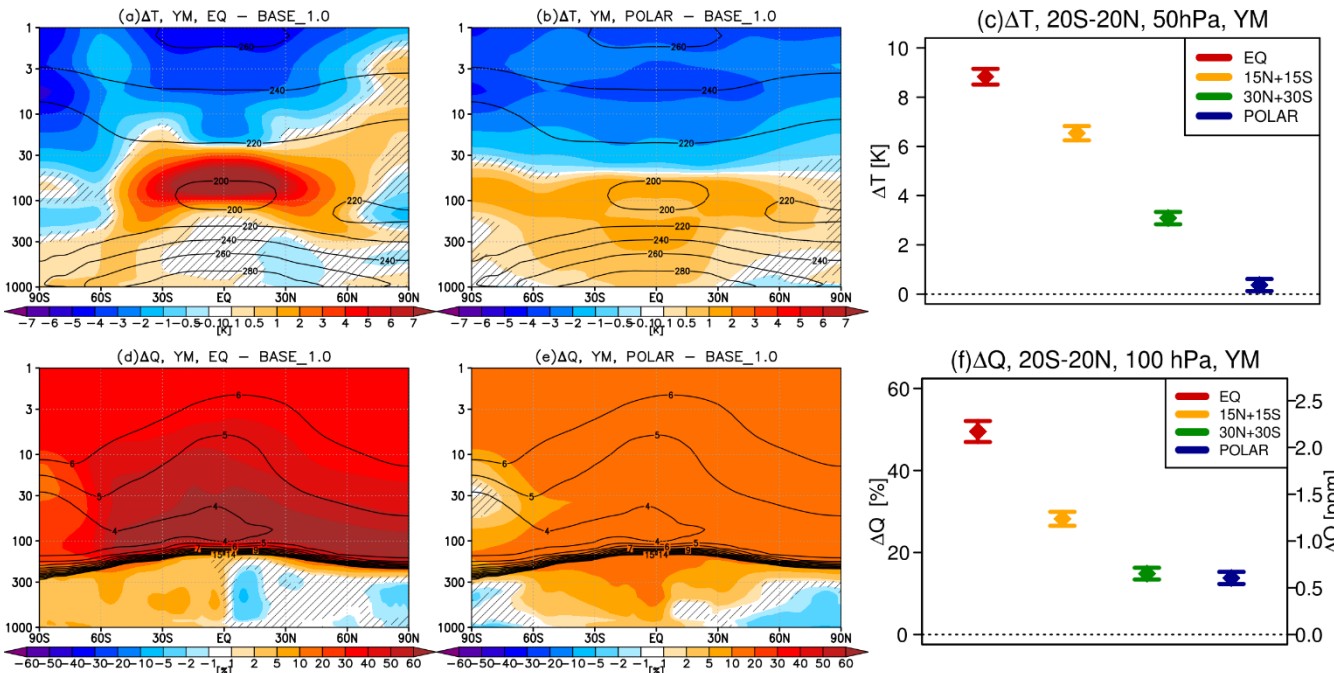

**Figure 2.** Shading (left and middle): yearly mean changes in (top) temperatures and (bottom) water vapour for (left) EQ and (middle) POLAR strategy compared to the baseline period (2008-2027). Contours show the corresponding values in the baseline period for refence (in units of K and ppm for temperature and water vapour respectively). Hatching indicate regions where the response is not statistically significant, taken as smaller than ±2 standard errors of the difference in means. See Fig. S2 in Supplementary Material for the analogous changes in the 15N+15S and 30N+30S strategies. Right columns show the changes in the tropical (20°S-20°N) temperatures at 50 hPa and water vapour at 100 hPa in each of the four strategies. The errorbars denote ±2 standard error of the difference in means.

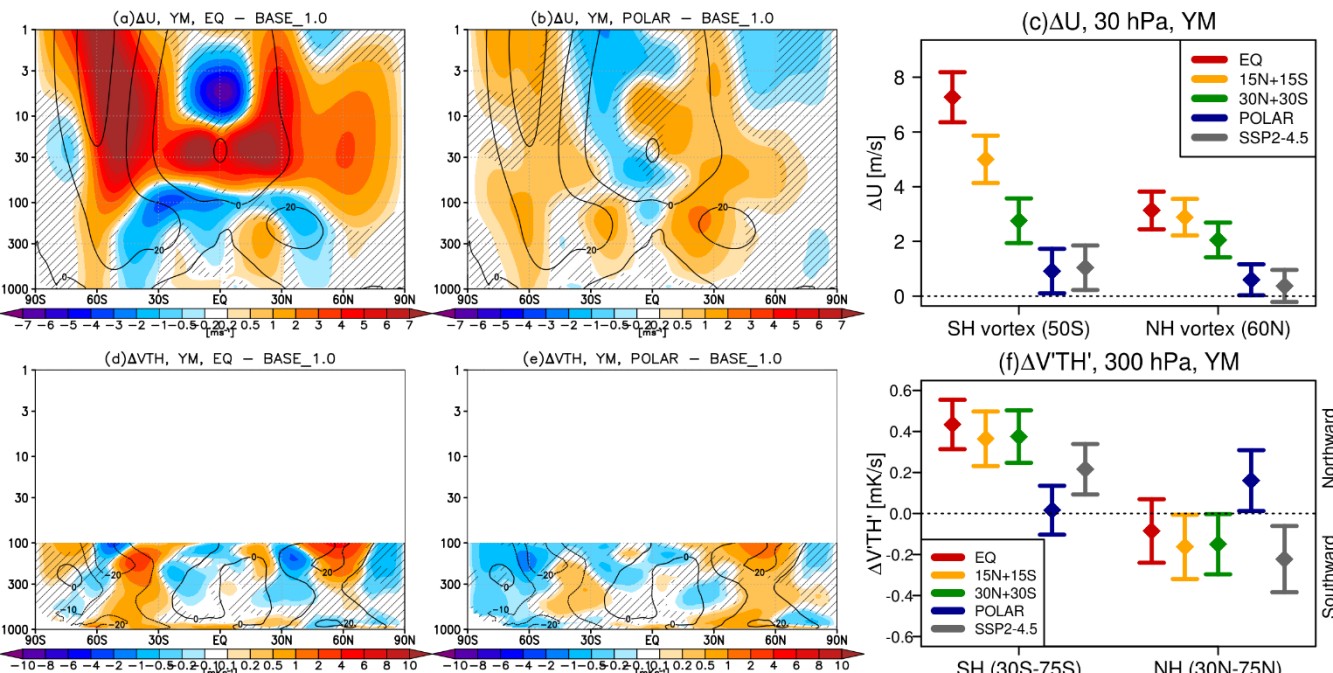

**Figure 3.** Shading (left and middle): yearly mean changes in (top) zonal winds and (bottom) northward meridional eddy heat flux below 100 hPa for (left) EQ and (middle) POLAR strategy compared to the baseline period. Contours show the corresponding values in the baseline period for refence. See Fig. S3 in Supplementary Material for the analogous changes in the 15N+15S and 30N+30S strategies. Right column shows the changes in (top) the strength of the NH (60°S) and SH (50°S) polar vortex at 30 hPa and (b) northward meridional heat flux at 300 hPa averaged over the NH (30°N-75°N) and SH (30°S-75°S) extra-tropics for each of the four strategies and the SSP2-4.5 scenario compared to the baseline period. Hatching in (left-middle) and errorbars in (right) as in Fig. 2.

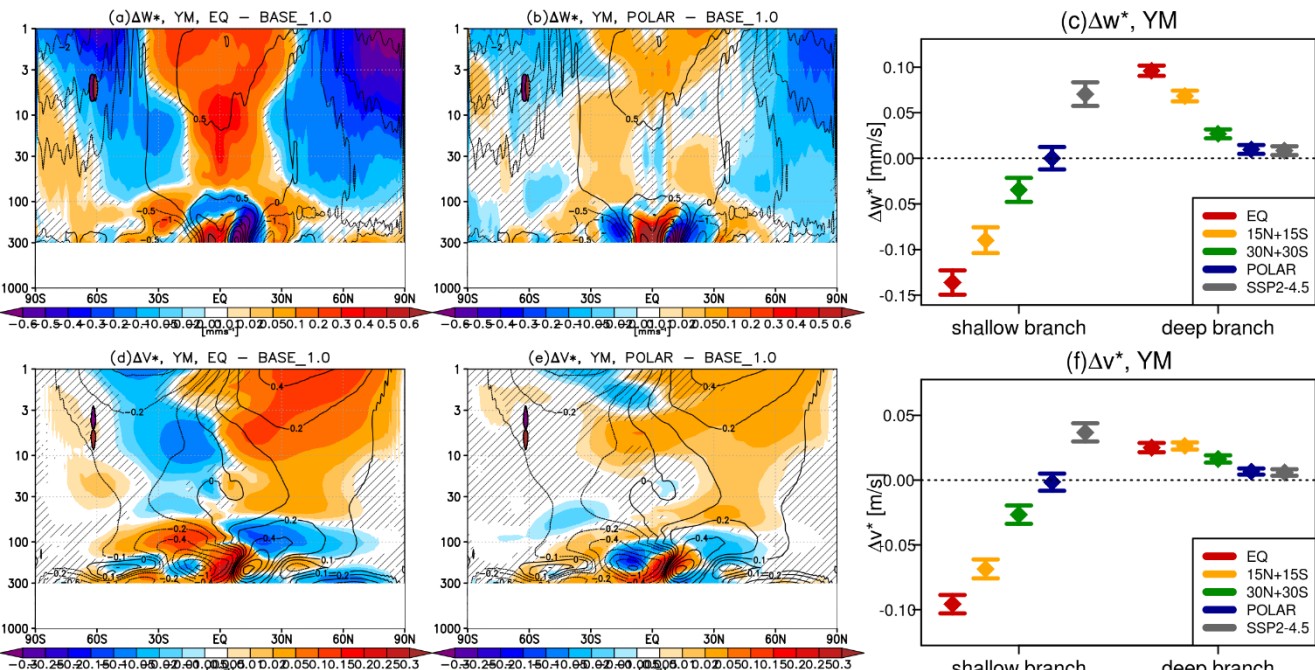

**Figure 4.** Shading (left and middle): yearly mean changes in residual (top) vertical and (bottom) meridional velocities for (left) EQ and (middle) POLAR strategy compared to the baseline period. Contours show the corresponding values in the baseline period for refence. See Fig. S7 in Supplementary Material for the analogous changes in the 15N+15S and 30N+30S strategies. Right column shows the changes in the shallow and deep branches of BDC; these are defined as means over 15°S-15°N at 100 hPa and over 25°S-25°N at 30 hPa, respectively, for w* and as difference between 0°-90°N and 90°S-0° at 70 hPa and at 30 hPa, respectively, for v*. Hatching in (left-middle) and errorbars in (right) as in Fig. 2.

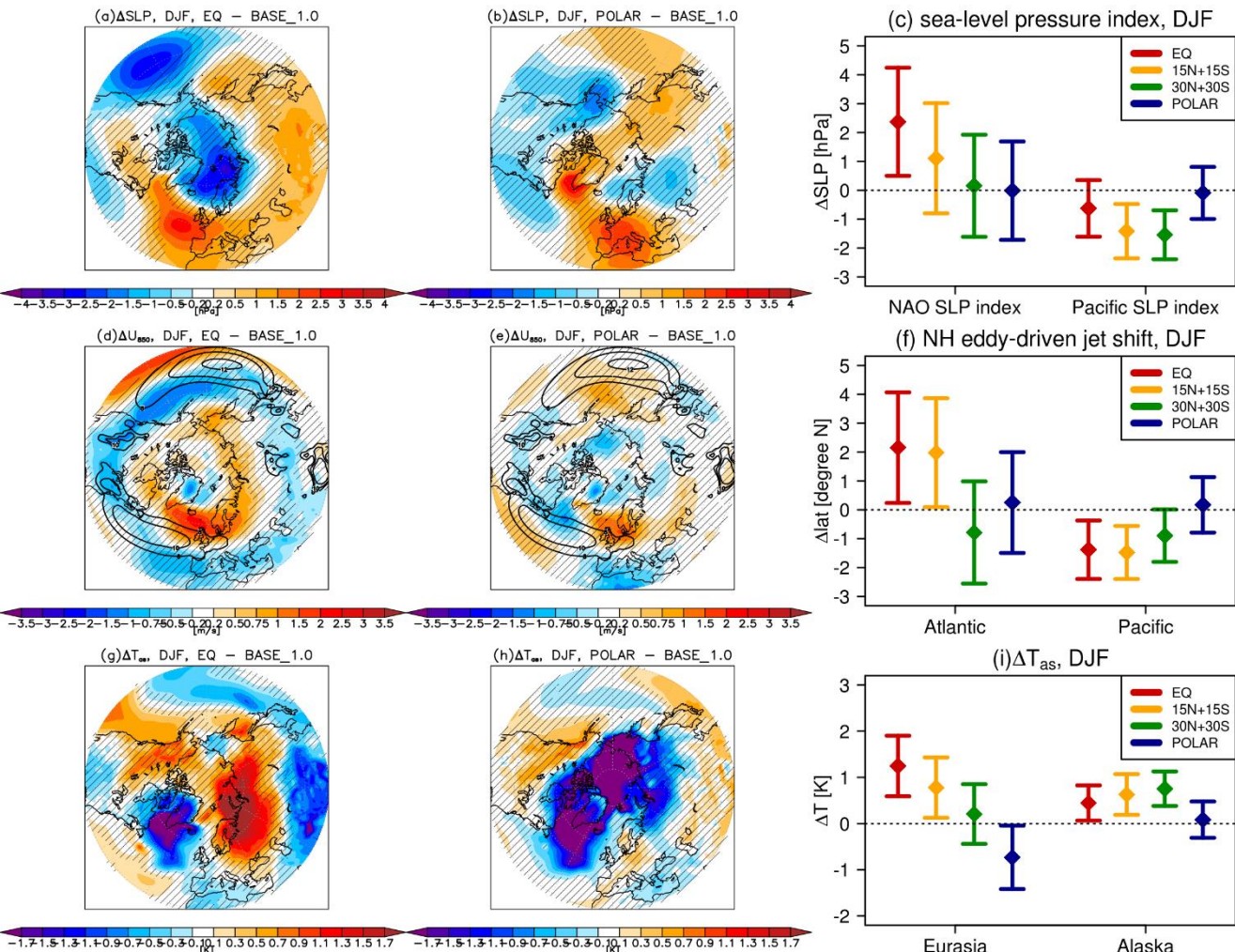

**Figure 5.** Shading (left and middle): DJF mean changes in (top) sea level pressure, (middle) zonal wind at 850 hPa, and (bottom) near-surface atmospheric temperatures northward from 30°N for (left) EQ and (middle) POLAR strategy compared to the baseline period. Contours in (middle) show the corresponding values in the baseline period for refence and denote the position of the climatological jet. See Fig. S9 in Supplementary Material for the analogous changes in the 15N+15S and 30N+30S strategies. Right columns show changes in the NAO and Pacific sea-level pressure index, position of the Atlantic and Pacific eddy-driven jet, and temperatures over Eurasian and Alaska regions (see text for details). Hatching in (left-middle) and errorbars in (right) as in Fig. 2.

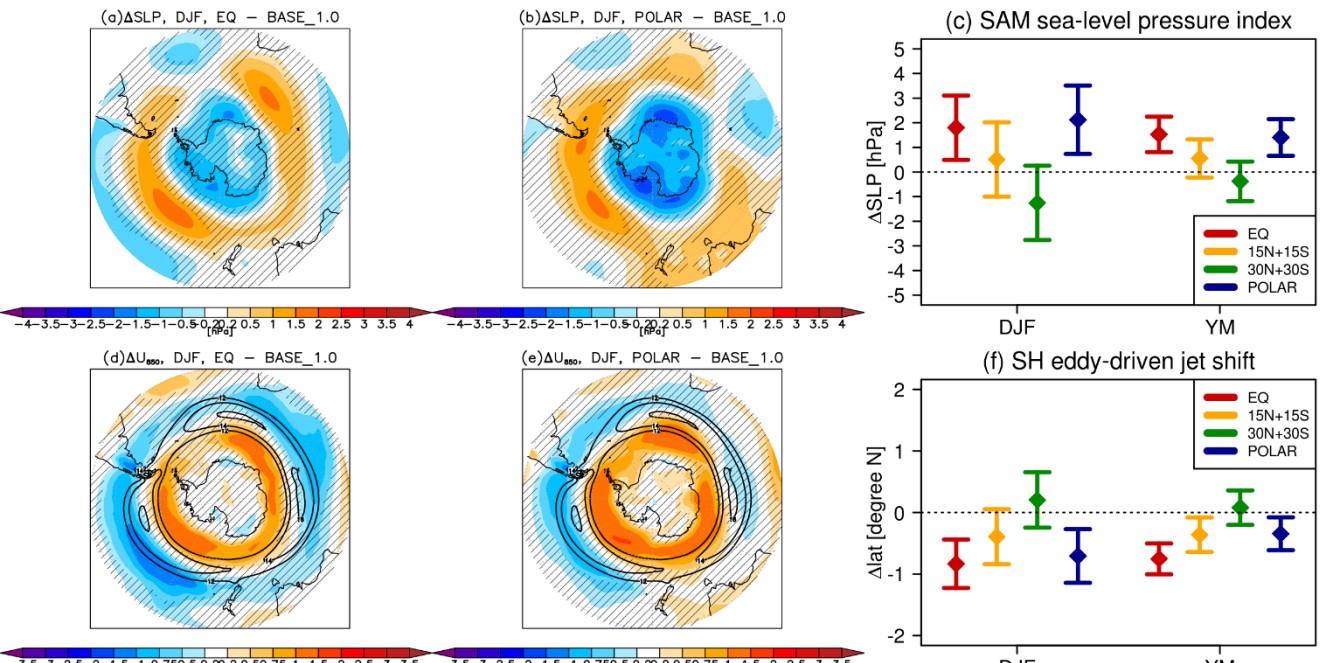

**Figure 6. Shading (left and middle): DJF mean changes in (top) sea level pressure, and (bottom) zonal wind at 850 hPa southward from 30°S for (left) EQ and (middle) POLAR strategy compared to the baseline period. Contours in (middle) show the corresponding values in the baseline period for refence and denote the position of the climatological jet. See Fig. S12 in Supplementary Material for the analogous changes in the 15N+15S and 30N+30S strategies. Right columns show changes in DJF mean and yearly mean SAM sea level pressure index and the position of SH eddy-driven jet (see text for details). Hatching in (left-middle) and errorbars in (right)**
**as in Fig. 2.**

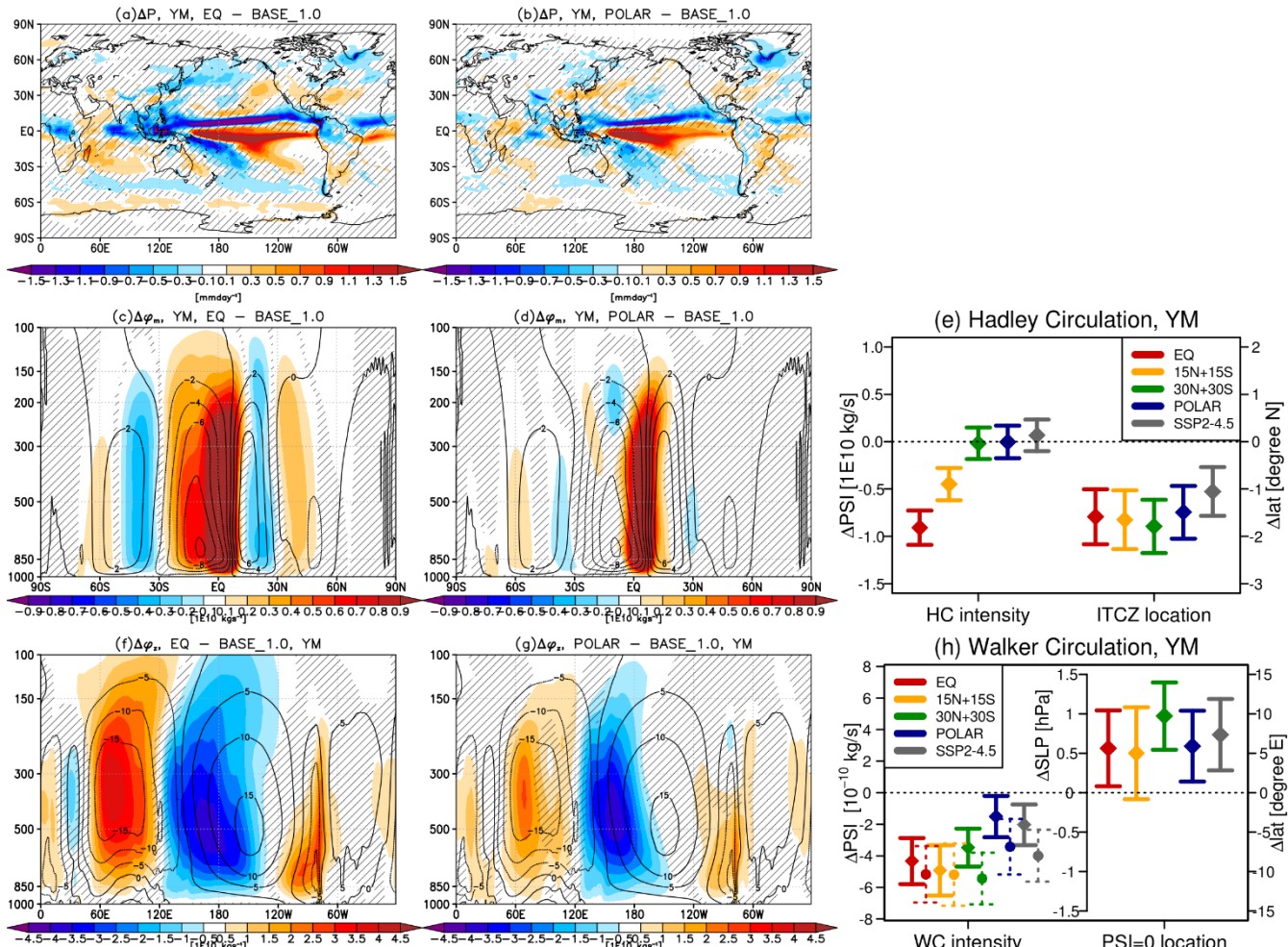

**Figure 7.** Shading (left and middle): Yearly mean changes in (top) precipitation, (middle) meridional mass stream function, and (bottom) zonal mass stream function for (left) EQ and (middle) POLAR strategy compared to the baseline period. Contours in middle and bottom show the corresponding values in the baseline period for refence. See Fig. S13 in Supplementary Material for the analogous changes in the 15N+15S and 30N+30S strategies. Right column shows changes in the intensity of Hadley Circulation and the position of ITCZ, and in the intensity of Walker Circulation (diamond and solid lines show the results derived using the stream function method, point and dashed lines show the results derived using the sea-level pressure method) and the position of the transition between its Pacific and Indian Ocean cells for each of the four SAI strategies as well as the control SSP2-4.5 simulation (see text for details). Hatching in (left-middle) and errorbars in (right) as in Fig. 2.

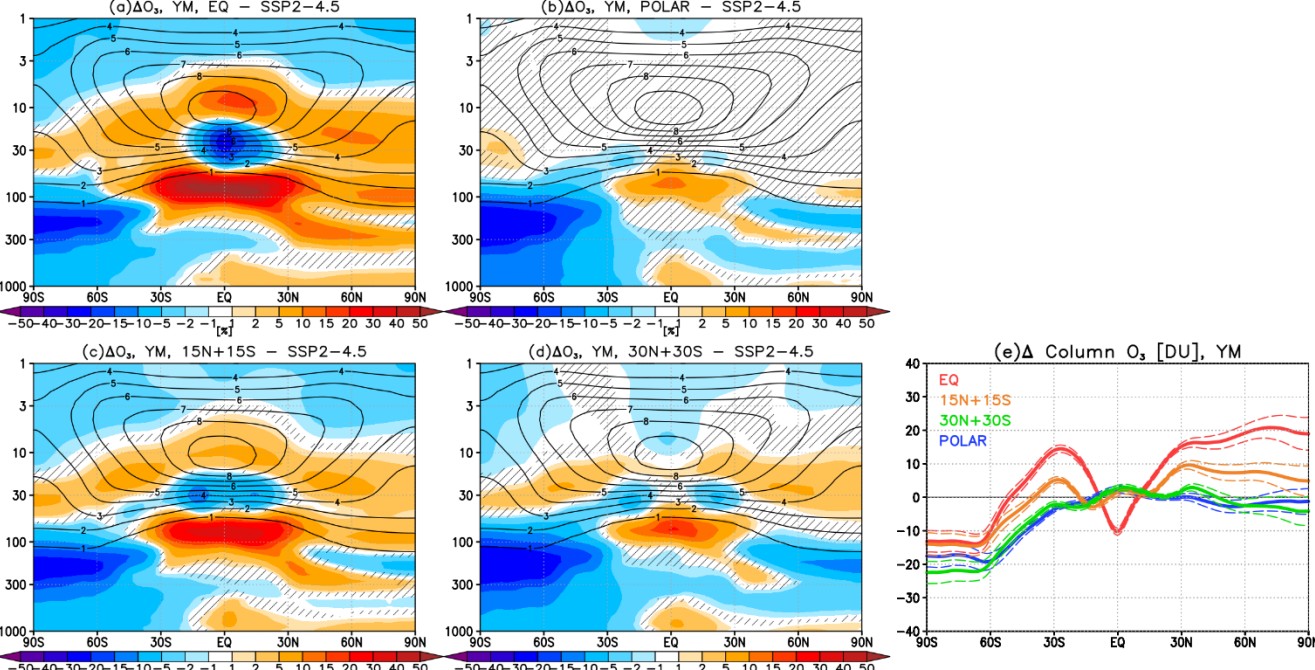

**Figure 8.** Shading (left and middle): Yearly mean changes in ozone mixing ratios for each of the EQ, 15N+15S, 30N+30S and POLAR strategies compared to the same period (i.e. 2050-2069) of the control SSP2-4.5 simulation. Contours show the corresponding values in SSP2-4.5 for refence (in the units of ppm). Hatching as in Fig. 2. Right: Yearly mean changes in total column ozone for each of the four strategies as a function of latitude. Dashed lines indicate the associates ±2 standard errors of the difference in means.

865

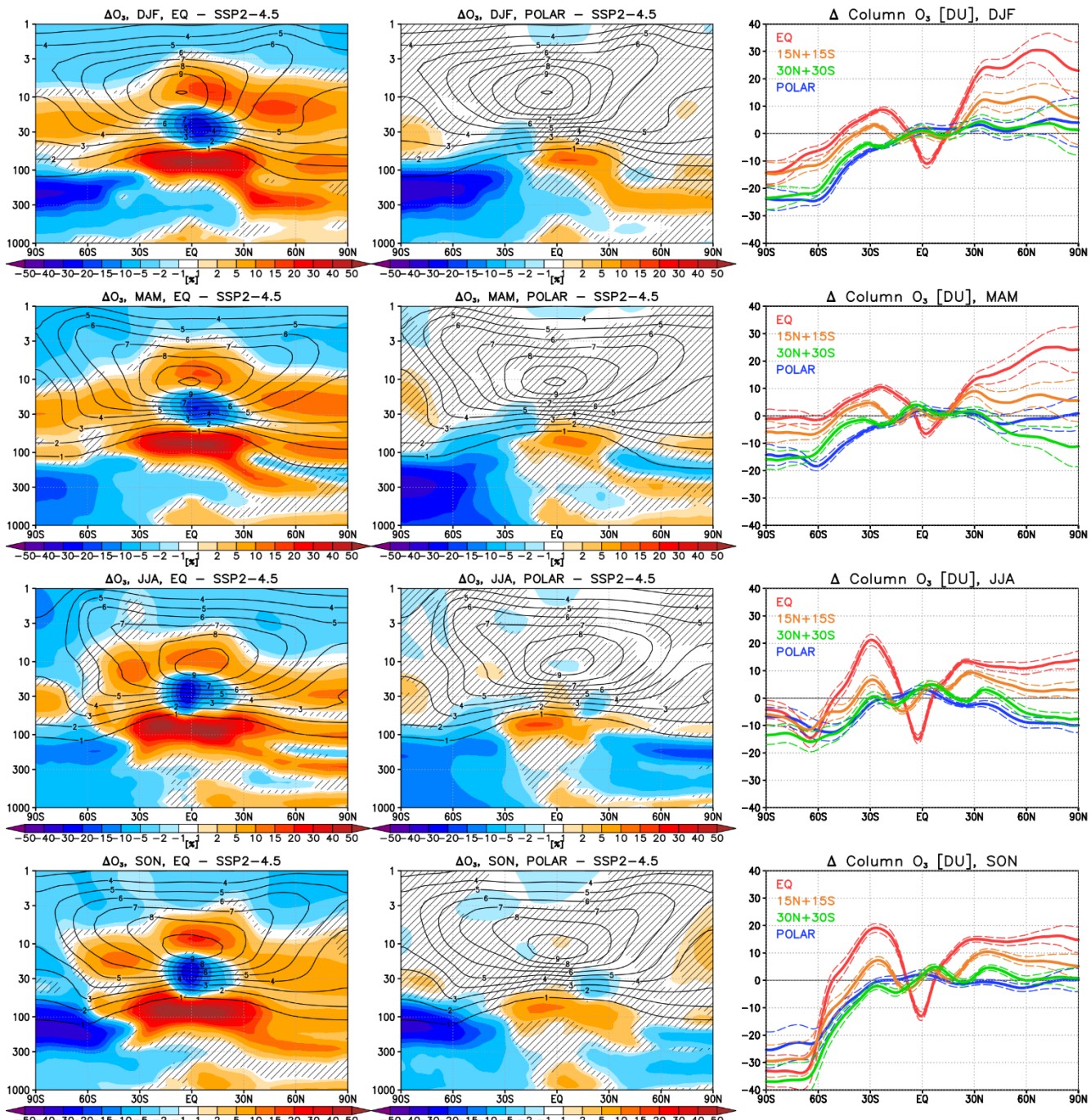

**Figure 9.** Shading (left and middle): Seasonal mean (top to bottom: DJF, MAM, JJA and SON) changes in ozone mixing ratios for the (left) EQ and (middle) POLAR strategies compared to the same period (i.e. 2050-2069) of the control SSP2-4.5 simulation. Contours show the corresponding values in SSP2-4.5 for refence (in the units of ppm). Hatching as in Fig. 2. See Fig. S15 in Supplementary Material for the analogous changes in the 15N+15S and 30N+30S strategies. Right: Seasonal mean changes in total

column ozone for each of the four strategies as a function of latitude. Dashed lines indicate the associated ±2 standard errors of the difference in means.