# Peer review of "Injection strategy - a driver of atmospheric circulation and ozone response to stratospheric aerosol geoengineering"

_EGUsphere, 2023_

## Referee Comment (RC1)

Review on:

**Injection strategy - a driver of atmospheric circulation and ozone response to stratospheric aerosol geoengineering**

Ewa Bendarz et al

April 20, 2023

The authors analyze the results of four simulations of different injection scenarios on stratospheric aerosol intervention (SAI), or stratospheric aerosol geoengineering as they say in the title. The area of injection changes from equator to mid latitudes. The simulations are partly new, especially the assumption for seasonally varying injections at 60N and 60S. The other injection strategies are similar to the GeoMip5 scenarios or previous publications. The authors analyses different impacts of SAI on stratospheric and tropospheric dynamics. The analysis of the simulations includes an interesting discussion on climate and tropospheric circulation, e.g. NAO. This important aspect has not been taken into account enough in existing literature. To me, this is the main aspect of the paper.

The paper is well written and reads well. I recommend publication after a few minor corrections.

Ulrike Niemeier

**General:**

As stated above, the impact of SAI on the tropospheric circulation is an important aspect in the discussion about SAI. Other parts, e.g. the impact on the Brewer Dobson Circulations, ozone or temperature have bee discussed earlier. Here the authors should cite broader and discuss their work in relation to previous publications. They have to state clearly which aspects of the analysis are new. GeoMIP 5 scenarios used RCP4.5 forcing as well. They showed a rather small signal to noise ration. Therefore, a short discussion why this paper bases SAI on RCP4.5 should be added.

You show mainly yearly averages in the main paper. The POLAR injections depend on season and have, therefore, very different seasonal aspects. This needs to be taken more into account.

I wonder a little about the title. A discussion on injection strategy is not new. The impact on high latitude tropical circulation and climate are far less discussed in previous literature. You use geoengineering in the title, but not again in the text. Stick to one of both.

**Specific comments:**

All Figure: Increase font size of the legend.

Introduction: Please put your work better into relation to previous work. Injection strategies have been discussed before. Why d we need another paper.

Methods:

Line 100: Generates the model a well developed QBO or more a QBO like pattern? The vertical resolution seems to me a bit low for a good QBO.

Line 104-105: SSP2-4.5 and the period 2035 to 2069 results in a low signal to noise ration. Why this scenario? The world in in 2023 on the 8.5 track.

Line 111: An injection altitude of 22 km is high. On the one hand, this kind of study aims a bit on better deployment strategies. On the other hand, the injection altitude would be difficult to do. So, why 22 km?

Do you have an ensemble or single simulations?

Line 120: Please, include Fig S1 into the main paper.

Line 124: POLAR... highest aerosols concentration... Where? Not in the annual mean.

3. Annual mean changes......

An annual mean cannot cover the impacts discussed here. The strategy of polar is not annual. I wondered a bit, if some impacts were hided behind the annual mean. Add seasonal means here, the paper will clearly gain.

Line 157: .. discussed in or detail in Zhang....... Please quantify and add a few sentences.

Line 170: Do I understand this right, your model cannot calculate the RF of sulfate? No radiation double call?

Line 178-179: Fig 2 does not show a weakening of the gradient. The isolines show 200 K in the topics and 220 in NH. This will not result in a stronger temperature gradient when warming the tropics. Change the plots and/or the discussion accordingly.

Line 180: Please be more precise. Which jets, where is the westerly response?

Line 180pp: This discussion is useless with plots of annual means. Add seasonal plots for the discussion of polar vortex, in case you mean polar vortex as you don't say so. Add seasonal plots in general, esp for POLAR.

Line 198: No enhanced gradient in your figure of temperature anomaly.

Line 236: Where do I see 15N+15S? Reference is missing.

Fig 4: TREFHT?

Fig 6: Precipitation changes between EQ and POLAR seem to be small and mainly over water. Changes over land might be more critical in POLAR.

Fig 5.2: Do we really get a good impression from yearly mean data? Changes under POLAR are strong as well and one may oversee important aspects this way.

Discussion:
This is mainly a summary. A discussion of the results is missing, e.g. which strategy may have stronger impact on land precipitation, monsoon (GeoMIP5 papers) etc. This is a single model study. There are many studies out to discuss shortly how much the results depend on the model.

Line 422: Temperature do not increase only in the tropical lower stratosphere.

Line 454: Bendarz(2022b): say a word about the content when cited here. The reader has to open the paper to follow you.

Please sort the reference list. Also, titles are missing.

---

## Author Comment (AC1)

**AUTHORS RESPONSE TO REVIEWER #1**

The authors analyze the results of four simulations of different injection scenarios on stratospheric aerosol intervention (SAI), or stratospheric aerosol geoengineering as they say in the title. The area of injection changes from equator to mid latitudes. The simulations are partly new, especially the assumption for seasonally varying injections at 60N and 60S. The other injection strategies are similar to the GeoMip5 scenarios or previous publications. The authors analyses different impacts of SAI on stratospheric and tropospheric dynamics. The analysis of the simulations includes an interesting discussion on climate and tropospheric circulation, e.g. NAO. This important aspect has not been taken into account enough in existing literature. To me, this is the main aspect of the paper. The paper is well written and reads well. I recommend publication after a few minor corrections. Ulrike Niemeier

We thank the reviewer for the positive review and helpful comments that have improved our manuscript. We address the specific comments below in blue.

**General:**

As stated above, the impact of SAI on the tropospheric circulation is an important aspect in the discussion about SAI. Other parts, e.g. the impact on the Brewer Dobson Circulations, ozone or temperature have bee discussed earlier. Here the authors should cite broader and discuss their work in relation to previous publications. They have to state clearly which aspects of the analysis are new.

We note that the main GeoMIP experiments always injected  $SO_2$  at the equator, or close by (Visioni et al., 2023, https://doi.org/10.5194/acp-23-5149-2023). While we acknowledge that other previous studies might have analysed some of the four strategies discussed here, none of them compared all of them in a consistent way, and most of them used fixed amounts of sulfur injections and an atmosphere-only model configuration.

Instead, one of the main advances of our study is that we consistently compare four injection strategies that result in similar global mean surface temperature response. Something similar to this was only done in Kravitz et al. (2019) with CESM1 but using only two injection strategies (an equatorial and a multi-latitude multi-objective strategy). We have now made sure we highlight the novelty of our study in the revised manuscript, as well as include more discussion with previous strategy exploration studies such as Franke et al (2021), Weisenstein et al (2022) and Laasko et al. (2022). We also note that we examine some SAI impacts on aspects of the climate system that previously have not been explored in detail in relation to injection strategy.

GeoMIP 5 scenarios used RCP4.5 forcing as well. They showed a rather small signal to noise ration. Therefore, a short discussion why this paper bases SAI on RCP4.5 should be added.

We have now added this – see the response to the specific comment below.

You show mainly yearly averages in the main paper. The POLAR injections depend on season and have, therefore, very different seasonal aspects. This needs to be taken more into account.

The reviewer is correct that it is important to consider not only annual means but also seasonal impacts, especially for changes to the polar vortex. We note that we do, however, already include seasonal mean zonal wind changes in the Supplement, and, we now also added the corresponding seasonal mean changes in temperature. The analysis of the impacts on the modes of extra-tropical modes of variability, i.e. NAM and SAM (Section 4.2 and 4.3), which are directly important for the

surface climate changes, is made using December-to-February mean data, i.e. the season when the contribution of top-down changes in the stratosphere is particularly important.

I wonder a little about the title. A discussion on injection strategy is not new. The impact on high latitude tropical circulation and climate are far less discussed in previous literature. You use geoengineering in the title, but not again in the text. Stick to one of both.

We note that we do use the word 'geoengineering' in the main text. The beginning of the introduction reads: "Stratospheric Aerosol Injection (SAI) is a proposed solar geoengineering method aimed at temporarily offsetting some of the negative impacts of rising greenhouse gas levels and the resulting increases in surface temperatures"

The choice of this word in the title was made to avoid using the word 'injection' twice.

**Specific comments:**

**All Figure:**

Increase font size of the legend.

**Done.**

**Introduction:**

Please put your work better into relation to previous work. Injection strategies have been discussed before. Why d we need another paper.

Thank you – as noted in the response to the general comment above, we have now made sure we highlight the novelty of our study in the revised manuscript, as well as include more discussion with previous strategy exploration studies such as Franke et al (2021), Weisenstein et al (2022) and Laasko et al. (2022).

**Methods:**

Line 100: Generates the model a well developed QBO or more a QBO like pattern? The vertical resolution seems to me a bit low for a good QBO.

The reviewer is correct in that the simulated QBO in this model version has some deficiencies compared to observations – we have now added more details about the characteristics of the model QBO to the text.

We note that while a CESM2(WACCM6) version with increased vertical resolution (110 levels instead of 70) exists and produces a more realistic QBO (Garcia and Richter, 2019, doi: https://doi.org/10.1175/JAS-D-18-0088.1), the version is substantially more computationally expensive that the standard 70-layer version used here. As such, we have decided to use the standard version as a trade-off between improved representation of the QBO and being able to simulate more SAI strategies.

**Line 104-105: SSP2-4.5 and the period 2035 to 2069 results in a low signal to noise ration. Why this scenario? The world in in 2023 on the 8.5 track.**

Our experiments were designed following the considerations outlined in MacMartin et al. (2022). As discussed in Burgess et al. (2020) and UNEP (2021), the SSP2-4.5 scenario is roughly consistent with the Paris Agreement's Nationally Determined Contributions without increased ambition. We note that in the short-term, most of the SSP scenarios look alike (and temperature is a function of the

cumulative emissions, not the instantaneous ones). Nonetheless, we do not plan nor claim to be predicting the future here (and explain our reasoning in depth in MacMartin et al. 2022), but have now expanded the discussion in the text to justify our decision better.

**Line 111: An injection altitude of 22 km is high. On the one hand, this kind of study aims a bit on better deployment strategies. On the other hand, the injection altitude would be difficult to do. So, why 22 km?**

The injection altitude is actually 21.5 km (we have clarified/changed this in the text). In general, the choice of injection altitude represents a trade-off between larger technological difficulties in case of real-world deployment for higher altitude injections and reduced cooling efficiency for lower altitude injections due to shorter aerosol lifetime and offsetting radiative impacts (as discussed in Lee et al., 2023). As discussed in MacMartin et al. (2022), the 21.5 injection altitude appears plausibly achievable with existing aircraft engines (Bingaman et al., 2020, https://doi.org/10.2514/6.2020-0618).

We note that the injection altitude of 21.5 km constitutes an improvement compared to the previous widely-used CESM1 GLENS SAI simulations, which injected SO2 roughly at 7 km above the tropical tropopause (Tilmes et al., 2018).

**Do you have an ensemble or single simulations?**

We apologize for forgetting to mention the number of ensemble members (i.e. 3) per strategy used – we have now added this information.

Line 120: Please, include Fig S1 into the main paper.

As suggested, we have now included it; although for consistency with the rest of the manuscript we include only the panels for the EQ and POLAR strategies in the main manuscript, with the remainder of the strategies in the Supplement.

Line 124: POLAR... highest aerosols concentration... Where? Not in the annual mean.

Yes, as shown in Fig. 1b,f. To be clear, we meant "maximum aerosol concentration" relative to other latitudes; we have changed the text.

3. Annual mean changes.....

An annual mean cannot cover the impacts discussed here. The strategy of polar is not annual. I wondered a bit, if some impacts were hided behind the annual mean. Add seasonal means here, the paper will clearly gain.

The reviewer is correct that it is important to consider not only annual means but also seasonal impacts, especially for changes to the polar vortex. We note that we do, however, already include seasonal mean zonal wind changes in the Supplement, and, we now also added the corresponding seasonal mean changes in temperature and point this out in Section 3. As discussed in the response to the corresponding comment below, we prefer to keep the seasonal mean changes in the supplement to avoid detailed large multi-panel figure and lengthy discussions of the contributing factors that may distract from the main conclusions. Further discussion of the seasonality of the zonal wind and temperature responses is found in Section 4.

Line 157: .. discussed in or detail in Zhang...... Please quantify and add a few sentences.

Done – the part now reads: "The particularly strong increase in lower stratospheric water vapour in EQ, up to 75% at 70 hPa, thus contributes to the low efficacy of this strategy (with 21 Tg-SO2/yr needed in EQ to reach the temperature target, compared to 14 and 16 Tg-SO2/yr in 30N+30S and 15N+15S, respectively; Section 2.1) that is also caused by the strong tropical confinement of aerosols and their large size (as discussed in more detail Section 2.2 here and in Zhang et al., 2023)."

**Line 170: Do I understand this right, your model cannot calculate the RF of sulfate? No radiation double call?**

We apologize for the confusion – the CESM model can indeed indirectly calculate the RF of sulfate using a double call to the radiative scheme (as it was done in the study of Visioni et al., 2022, the result of which we use here), but the diagnostics needed were unfortunately not outputted in our simulations.

Line 178-179: Fig 2 does not show a weakening of the gradient. The isolines show 200 K in the topics and 220 in NH. This will not result in a stronger temperature gradient when warming the tropics. Change the plots and/or the discussion accordingly.

We realize we were not clear here that we were referring to the SAI responses (anomalies), and not the total changes. We have rephrased this to "The SAI-induced warming in the tropical lower stratosphere drives an anomalous strengthening of the equator-to-pole meridional temperature gradients near the tropopause and lower stratosphere. This drives an anomalous increase of the subtropical to extratropical stratospheric westerly winds in both hemispheres via thermal wind balance in all seasons and most injection strategies, though more intermittently for the seasonal injection in POLAR (Fig. S4-S5). In the winter and spring hemisphere, especially in the NH, the strengthening of the polar stratospheric jet at ~60° latitude is likely the result of the associated modulation of atmospheric wave propagation and convergence due to the more westerly subtropical winds (Fig. S5; see also e.g. Walz et al., 2023)"

Line 180: Please be more precise. Which jets, where is the westerly response?

We have now clarified this; see the response above.

Line 180pp: This discussion is useless with plots of annual means. Add seasonal plots for the discussion of polar vortex, in case you mean polar vortex as you don't say so. Add seasonal plots in general, esp for POLAR.

We agree with the reviewer that the seasonal mean plots are useful when discussing the behaviour of the polar vortex. We note that we do, however, show the seasonal mean zonal wind changes in the Supplementary material, and we refer to them in Section 3.3.1. See also our response above. Note that the *responses* to SAI are similar across seasons, whether the polar vortex is present or not (we have now made sure this is clear in the text, as there are westerly wind responses even in the summer hemisphere with no polar vortex).

Additionally, we have decided to focus on the yearly mean responses in the main manuscript as the analysis is performed for the SAI in the future period minus quasi present day baseline period, and as such detailed assessment of the seasonal wind changes in these SAI strategies needs to take into account not only the SAI-induced impacts, but also long-term changes in other circulation drivers, especially long-term ozone recovery from the reduction in ozone depleting substances; the latter plays a particularly important role during SH spring and summer (Fig. R1 below). As such we prefer to keep the analysis of the seasonal mean responses in the supplement; however, we have now included the corresponding seasonal mean temperature responses to the supplement, and to assist

interpretation of the derived changes, we have also added yearly mean zonal wind responses simulated in SSP2-4.5 to Fig. 3c,f for reference.

**Line 198: No enhanced gradient in your figure of temperature anomaly.**

This text has now been rewritten for clarity.

Line 236: Where do I see 15N+15S? Reference is missing.

Thank you for spotting this – we have now added the missing figure references.

Fig 4: TREFHT?

Corrected – we meant 'Tas'.

Fig 6: Precipitation changes between EQ and POLAR seem to be small and mainly over water. Changes over land might be more critical in POLAR.

We are not sure what the reviewer means – precipitation responses in general tend to be small and/or not statistically significant, but here all SAI strategies give rise to some precipitation responses over both land and ocean.

Fig 5.2: Do we really get a good impression from yearly mean data? Changes under POLAR are strong as well and one may oversee important aspects this way.

Figure 5 shows the responses simulated for December-to-February mean (i.e. when the contribution of top-down changes in the stratosphere is particularly important), and this is compared with the yearly mean responses in panels c and f. The comparison yields very similar responses in both cases. Analysis of the responses in other seasons (Fig. R2 below) shows overall similar responses.

**Discussion:**

This is mainly a summary. A discussion of the results is missing, e.g. which strategy may have stronger impact on land precipitation, monsoon (GeoMIP5 papers) etc. This is a single model study. There are many studies out to discuss shortly how much the results depend on the model.

The reviewer is right in highlighting that results are model dependent. However, while that has been discussed elsewhere, it was always in the context of one strategy for many models. Here, we are discussing strategy differences in one model, which is an important first step before discussing multiple strategies in many different models (which would also be unfeasible). We note that we highlight the need to test these one-model results, and the associated uncertainties, in a multi-model framework in the last paragraph of the summary/conclusion Section 7.

We have also now expanded it to include additional discussion about what might be considered when choosing an injection strategy in terms of impacts, i.e. the need to consider the variety of impacts when evaluating which strategy is most optimal: "We have demonstrated that some of the undesirable side-effects of SAI that have been well established for tropical injections - e.g. strengthening of the NH polar vortex and the resulting positive NAO-like surface response in winter, or weakening of the intensity of the Hadley and Walker circulations - appear to be mitigated for extra-tropical and polar injections. However, additional impacts for these strategies, like enhanced halogen activation on sulfate, changes to SAM or strengthening of the large-scale equator-to-pole gradient in case of the latter (see Fig. 1c in Zhang et al. 2023), need to also be considered, highlighting the complexity and trade-offs in evaluating which strategy is most optimal."

Line 422: Temperature do not increase only in the tropical lower stratosphere.

**Corrected.**

Line 454: Bendarz(2022b): say a word about the content when cited here. The reader has to open the paper to follow you.

We note that we already explain the content of Bednarz et al. (2022b), and it's relevance for the results in this manuscript, in Section 4.2, though we clarify the wording as shown below::

"Bednarz et al. (2022b) analysed the SAM changes under fixed single point SO2 injections imposed between 30°S and 30°N in the same CESM2 version, and showed that the SAM response becomes negative under SO2 injections in the SH as the injections are moved further into the subtropics. That work suggested that this occurs because of the poleward extent of lower stratospheric heating impacting planetary wave propagation in the stratosphere as well as eddy heat and momentum fluxes in the troposphere below. It is thus plausible that the SAM and jet responses in the EQ, 15N+15S and 30N+30S strategies here are largely dynamically driven by the lower stratospheric heating, in a manner consistent with Bednarz et al. (2022b)."

Please sort the reference list. Also, titles are missing.

Done.